# Seroprevalence of SARS-CoV-2 antibodies among Forcibly Displaced Myanmar Nationals in Cox's Bazar, Bangladesh 2020: a population-based cross-sectional study

Mahbubur Rahman [1], Samsad Rabbani Khan,[1] A S M Alamgir,[2]
David S Kennedy,[3] Ferdous Hakim [4], Egmond Samir Evers,[3] Nawroz Afreen [1],
Ahmed Nawsher Alam,[2] Md Sahidul Islam [4], Debashish Paul,[3] Rijwan Bhuiyan,[5]
Raisul Islam,[3] Adneen Moureen,[6] M Salimuzzaman,[7] Mallick Masum Billah,[1]
Ahmed Raihan Sharif,[1] Mst Khaleda Akter,[4] Sharmin Sultana,[2]
Manjur Hossain Khan,[2] Kai von Harbou,[3] Mohammad Mostafa Zaman [4],
Tahmina Shirin,[8] Meerjady Sabrina Flora[9]

For numbered affiliations see end of article.

**Correspondence to**
Dr Mahbubur Rahman;
dr_mahbub@yahoo.com

## ABSTRACT

**Objectives** The study aimed to determine the seroprevalence, the fraction of asymptomatic infections, and risk factors of SARS-CoV-2 infections among the Forcibly Displaced Myanmar Nationals (FDMNs).

**Design** It was a population-based two-stage cross-sectional study at the level of households.

**Setting** The study was conducted in December 2020 among household members of the FDMN population living in the 34 camps of Ukhia and Teknaf Upazila of Cox's Bazar district in Bangladesh.

**Participants** Among 860 697 FDMNs residing in 187 517 households, 3446 were recruited for the study. One individual aged 1 year or older was randomly selected from each targeted household.

**Primary and secondary outcome measures** Blood samples from respondents were tested for total antibodies for SARS-CoV-2 using Wantai ELISA kits, and later positive samples were validated by Kantaro kits.

**Results** More than half (55.3%) of the respondents were females, aged 23 median (IQR 14–35) years and more than half (58.4%) had no formal education. Overall, 2090 of 3446 study participants tested positive for SARS-CoV-2 antibody. The weighted and test adjusted seroprevalence (95% CI) was 48.3% (45.3% to 51.4%), which did not differ by the sexes. Children (aged 1–17 years) had a significantly lower seroprevalence 38.6% (95% CI 33.8% to 43.4%) compared with adults (58.1%, 95% CI 55.2% to 61.1%). Almost half (45.7%, 95% CI 41.9% to 49.5%) of seropositive individuals reported no relevant symptoms since March 2020. Antibody seroprevalence was higher in those with any comorbidity (57.8%, 95% CI 50.4% to 64.5%) than those without (47.2%, 95% CI 43.9% to 50.4%). Multivariate logistic regression analysis of all subjects identified increasing age and education as risk factors for seropositivity. In children (≤17 years), only age was significantly associated with the infection.

## STRENGTHS AND LIMITATIONS OF THIS STUDY

⇒ This study is one of the largest serosurveys in COVID-19 pandemic among displaced/refugee population.

⇒ Two-stage population-based design of the study with households identified randomly from the Family Counting Number using probability proportionate to size of the camps and participants recruited randomly from each selected households with a balanced number of males and females, ensured generalisability and gender equity in data collection.

⇒ Seroprevalence estimates were adjusted to account for test sensitivity and specificity of Wantai test kit validated by Kantaro to address issue of cross-reactivity.

⇒ The household roster could not be completed for all of targeted subjects because the listing of households was done several months before the survey. Non-response at the household level was higher than expected. This might have caused overestimation of the seroprevalence, which was addressed by population and household level weighting.

⇒ The study outcome relied on testing of antibody response, estimates of which might be affected by the waning antibody levels over time and the Wantai ELISA might have cross-reactivity with antibodies to other common human coronaviruses, leading to a false-positive result in some instances.

**Conclusions** In December 2020, about half of the FDMNs had antibodies against SARS-CoV-2, including those who reported no history of symptoms. Periodic serosurveys are necessary to recommend appropriate public health measures to limit transmission.

## INTRODUCTION

A novel pathogen's epidemiological and serological characteristics at the beginning of a pandemic remain uncertain. After the initial detection of SARS-CoV-2 in Wuhan of China in December 2019,[1] surveillance has focused primarily on clinical presentations and severity of COVID-19 disease. A full spectrum of the disease was not evident[2 3] Real-time reverse transcriptase-PCR (rRT-PCR) testing of people can identify patients, including those with mild symptoms. However, previous SARS-CoV-2 infections can be identified using antibody tests, including those who recovered or were not tested being asymptomatic.[4]

The global and regional data on the seroprevalence of antibodies against SARS-CoV-2 ranged during 2020 from 1.4% to 26.0% in countries, such as the USA,[5 6] Spain,[7] France,[8] Pakistan,[9] India[10] and Mongolia.[11] The first case of COVID-19 was reported in the Bangladeshi population on 8 March 2020, and the first case in Forcibly Displaced Myanmar Nationals (FDMNs) in Cox's Bazar district on 15 May 2020.[12] One study among Bangladeshi population between April and October 2020[13] estimated the national seroprevalence of IgG and IgM to be 30.4% and 39.7%, respectively. In Dhaka, the seroprevalence of IgG was 35.4% in non-slum areas and 63.5% in slum areas. In areas outside of Dhaka, the seroprevalence of IgG was 37.5% in urban areas and 28.7% in rural areas. Another study in Bangladesh reported a considerable variation of antibodies in symptomatic and asymptomatic COVID-19 patients.[14] However, seroprevalence estimates in FDMNs are absent, and studies documenting the seroprevalence of SARS-CoV-2 antibodies among displaced populations are limited globally. Bangladesh currently hosts around one million FDMNs (commonly known as 'Rohingyas', an ethnic group in the Rakhine State of Myanmar who are forcibly displaced in Bangladesh)[15] in two subdistricts of a south-eastern frontier district, Cox's Bazar, of Bangladesh.[16]

As of 31 March 2022, 5902 COVID-19 confirmed cases were reported from the FDMN camps with 42 deaths.[17] These camps are at high risk for COVID-19 due to their very high population density, limited access to improved sanitation and difficulties in implementing infection control measures.[18] Several outbreaks of communicable diseases have occurred in these camps in the past,[19] including measles, diphtheria and acute watery diarrhoea.

The aim of this survey, which has been adapted from the WHO Unity Study Protocol,[4] was to estimate the seroprevalence of SARS-CoV-2 antibodies and the proportion of asymptomatic infections among the people living in the FDMN camps in Cox's Bazar district of Bangladesh.

## METHODS
### Setting

We conducted this population-based two-stage cross-sectional survey in December 2020. The study was done on household members of the FDMN population using an adapted version of the WHO Unity Study Protocol.[4] All 34 FDMN camps of Cox's Bazar district were included (online supplemental figure 1), which present with a population density of 40 000 people/km$^2$ on average with some places reaching 70 000.[20] The sampling frame consisted of 860 697 FDMNs residing in 187 517 households registered with the Refugee Relief and Repatriation Commissioner (RRRC)-United Nations High Commissioner for Refugees (UNHCR) database as part of the RRRC-UNHCR Family Counting Exercise. UNHCR updates its FDMN database periodically collaboratively with the Government of Bangladesh.[21]

The sample size was calculated using an online tool for calculating proportion (OpenEpi).[22] Considering a hypothesised seroprevalence of 2.8%,[5] confidence limits±0.6%, design effect (1.5) and non-response adjustment (30%), the estimated minimum sample size was 6202.

In each camp, households were identified randomly by the Family Counting Number (FCN) using probability proportionate to size. Each identified household was assigned either as male or as female to ensure a balanced sample of males and females. One individual aged 1 year or more was selected from each identified household using a random number generator. Informed written consent was collected from the recruited respondents or their parents or legal guardians in case of children. Written assent was obtained from children aged 12–17 years. People who had any contraindication to venepuncture, such as bruising, swelling, infection or broken skin, were excluded from the study (figure 1).

### Questionnaire and data collection tool

The questionnaire was based on the WHO Unity Protocol[4] and included questions on demographic characteristics and symptom history. The questionnaire was translated into Bangla and the common language used in FDMN camps (Burmese) for participants' better understanding. Pretesting of the questions was carried out and the questionnaire was programmed using KoBo Toolbox[23] running on Android phones and tablets. Phones and tablets were password protected, and collected data were transferred at the end of each day of the activity.

### Field teams and training

The field teams consisted of doctors (team lead), nurses, laboratory technologists, paramedics and enumerators who were proficient in the local dialect. Forty-nine teams were deployed to collect data. Each field team comprised of two data collectors and one enumerator of the same sex. Female teams collected samples from households assigned as female and male teams collected samples from households assigned as male. In addition, community health workers/community leaders (Majhis) provided support to the teams to locate the selected households. The field teams' training included interactive discussions on all questions, mock interviews and blood sample collection exercises.

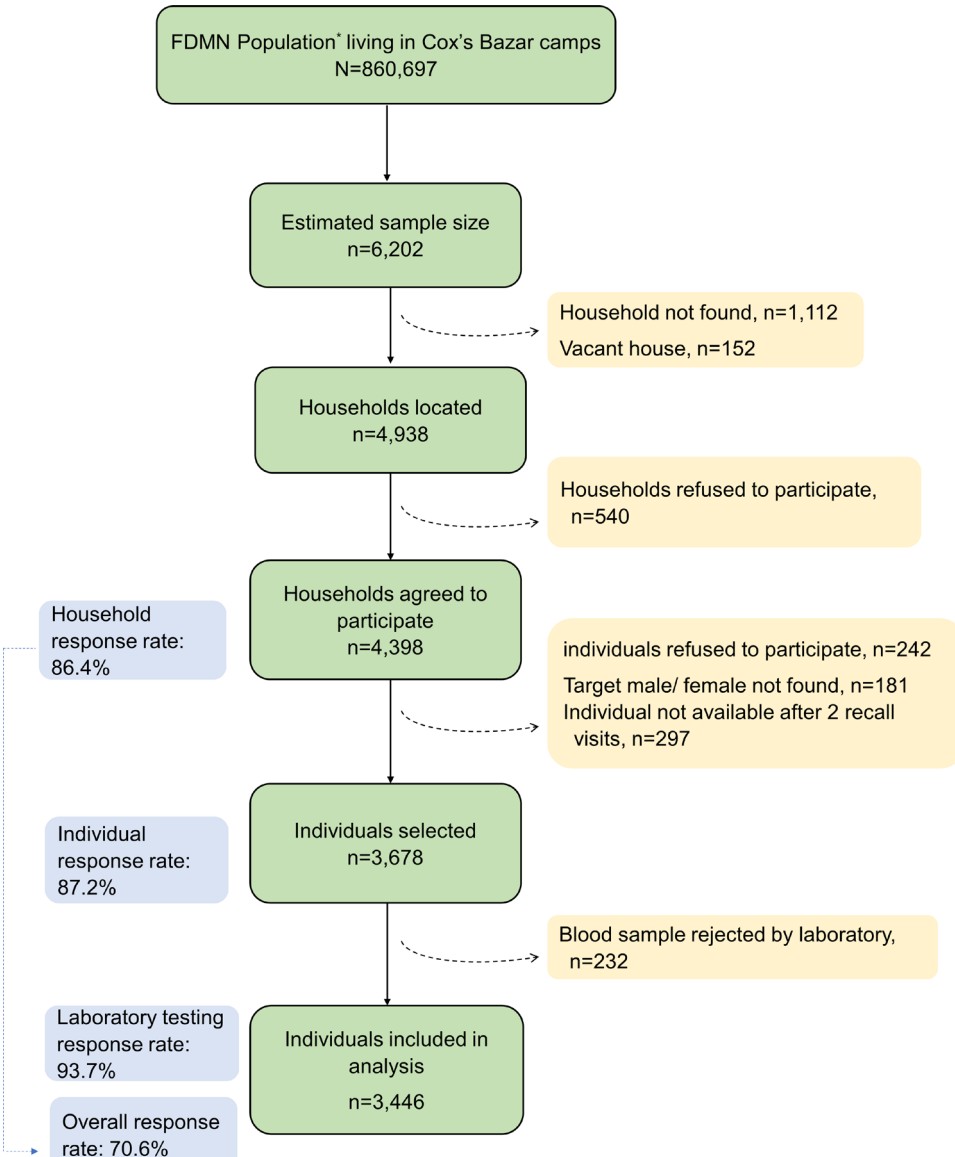

**Figure 1** Flow diagram showing selection of respondents of the FDMN population, Bangladesh December 2020. *Estimated from FDMN population frame of Refugee Relief and Repatriation Commissioner & United Nations High Commissioner for Refuge. FDMN, Forcibly Displaced Myanmar Nationals.

## Campaign to sensitise

Sensitisation meetings in all 34 FDMN camps were arranged before initiating data collection by the Camp In-charge assigned by the Government. Support from local religious leaders (eg, imam), Majhis, site management staff and community health workers of relevant partner organisations were sought to conduct the study. A flipchart developed by WHO was used to aid these sensitisation meetings.

## Specimen collection, preservation and testing procedures

A 5 mL (3 mL for young children) of venous blood was collected with aseptic precautions from each of the respondents. The samples were transported to specimen collection hubs maintaining proper labelling and temperature (2–8°C). At the hubs (one for each Upazila), the samples were checked for haemolysis, lipaemia and

inadequate quantity and sorted and sent to the IEDCR field laboratory at Cox's Bazar Medical College, Cox's Bazar for centrifugation at 3000 rpm for 20 min and labelled appropriately in aliquots (3 for each individual) for preservation at −80°C.

Samples with laboratory identification numbers were screened for the presence of SARS-CoV-2 virus-specific antibodies using Wantai SARS-CoV-2 Antibody ELISA kit[24] according to the manufacturer's instructions. One of the three aliquots was used for ELISA, and the remaining aliquots were stored at −80°C for further testing and quality control.

The blood samples were tested for the qualitative detection of total antibodies (IgG and IgM) to SARS-CoV-2 using a two-step incubation antigen 'sandwich' enzyme immunoassay where the receptor-binding domain of

SARS-CoV-2 spike protein was present. The patient's serum specimen was added during the first incubation, and a second recombinant SARS-CoV-2 antigen was added during the second incubation, respectively. The microwells were then washed to remove unbound conjugate, and chromogen solutions were added to identify the amount of antibody captured inside the well. Wells containing specimens negative for SARS-CoV-2 antibodies remained colourless.

The positive test results were validated by Kantaro test kits[25] which is a two-phase ELISA optimised for accurate detection of SARS-CoV-2 IgG antibodies, present in human serum samples with a specificity of 99.8% and a sensitivity of 97.8%.

### Quality assurance measures

The data collection procedure and teams were supervised by trained physicians. Independent monitoring visits were made by the investigators and other officials familiar with the study protocol. For external validation and quality control, 5% of the blood samples (positive and negative) were retested using Wantai kits at the IEDCR laboratory in Dhaka, the reference laboratory for SARS-CoV-2 in Bangladesh. Again, positive test results for antibodies to SARS-CoV-2 by Wantai test kits were validated by Kantaro test kits at the IEDCR reference laboratory.

### Data weighting and analysis

Data were downloaded from the cloud and merged with laboratory test results with unique identification numbers of respondents. After cleaning the dataset, weighting for FDMN population[21] was done following the method described by Hakim *et al*.[26] Base weight was calculated using a probability of selection of respondents among the eligible number of members of the household in a cluster defined by 34 camps, 5 age groups and 2 sexes. Later, base weight was adjusted with non-response weights separately for males and females. The final weight was generated after calibration to frame the population[21] in domains by camp, sex and age groups. Analysis was done using the final weights.

Categorical variables were reported as numbers and percentages, and continuous variables by median (IQR) or mean (SD) as appropriate. $\chi^2$ was used to test the independence between seropositive and categorical factors.

Our estimation of the seroprevalence of SARS-CoV-2 proceeded in two steps. First, we estimated the weighted seroprevalence of SARS-CoV-2 to represent the FDMN population. Later we used the Rogan-Gladen formula to adjust seroprevalence estimates to account for test sensitivity and specificity of Wantai validated by Kantaro.[27] The 95% CIs for the test-adjusted estimates were derived using bootstrap sampling, with 1 000 000 parametric bootstrap samples for each estimate, using the 'adjPrevSensSpecCI' function of the 'bootComb' R package.[28] More details on our bootstrap procedures are available in online supplemental material.

ORs and their 95% CIs were obtained using logistic regression analysis to identify the factors associated with weighted test adjusted seroprevalence. Multiple logistic regression models were applied on weighted data for children (1–17 years), adults (18 years and above) and all subjects combined. All variables were entered simultaneously into the multiple regression models. However, smoking could be used for adults only because smoking data were not collected for children. Alpha was set at 5% for considering statistical significance. All analyses were conducted using IBM SPSS V.26.0 and R V.4.1.1.

### Patient and public involvement

Sensitisation meetings were conducted in all FDMN camps before initiating data collection to ensure public involvement. Local religious leaders (eg, imam) and Majhis were involved to implement the study, especially for locating the households.

Patients or the public were not involved in the study design, or reporting, or dissemination plans.

## RESULTS

Among the estimated sample size of 6202, the study teams were successful in locating 4398 households yielding a response rate of 86.4%. Among the selected households, 3678 respondents (response rate, 87.2%) completed the individual interview and provided blood samples. Among the collected blood samples, 232 were rejected in the laboratory. Therefore, blood samples from 3446 respondents completed the laboratory test leading to a response rate of 93.7%. Data from these 3446 respondents with completed laboratory tests were used for the final analysis. Based on these three response rates, the overall response rate was 70.6% (online supplemental table 1). All results presented are based on weighted data.

### Background information of the study subjects

More than half of the respondents (55.3%) were female. More than two-thirds (69.4%) of them were aged 18 years or above. The median age (IQR) of all participants was 23 (14–35) years, which was not different by the sexes. More than half of the school eligible (age≥6 years) population reported no formal schooling (58.4%). Most of the respondents reported receiving BCG vaccination (86.3%). Nearly half (42.8%) of adult males (18 years and above) reported smoking tobacco, compared with 7.2% of females. About one in eight (12.5%) reported at least one comorbid condition. More than one-third (37.0%) of the respondents reported having at least one relevant symptom of COVID-19 disease since March 2020. The symptoms included fever, cough, dyspnoea, anosmia, chills, fatigue, sore throat, headache, ageusia, nausea, vomiting, diarrhoea, wheezing, chest pain, rhinorrhoea, conjunctivitis, other respiratory symptoms, muscle aches, myalgia, rash, loss of appetite, nosebleed and seizures (table 1).

**Table 1** Sociodemographic and health-related background information of study participants, FDMN population, Bangladesh December 2020 (N=3446)

| Variable | Overall n (%) | Female n (%) | Male n (%) |
|---|---|---|---|
| Age, years | | | |
| All | 3446 (100.0) | 1906 (100.0) | 1540 (100.0) |
| 1–17 | 1054 (30.6) | 472 (24.8) | 582 (37.8) |
| 18–94 | 2392 (69.4) | 1434 (75.2) | 958 (62.2) |
| Median (IQR) | 23.0 (14.0–35.0) | 24.0 (18.0–36.0) | 23.0 (11.0–35.0) |
| Education (n=3160)* | | | |
| No formal education | 1847 (58.4) | 1262 (70.5) | 585 (42.7) |
| Any schooling | 1313 (41.6) | 528 (29.5) | 785 (57.3) |
| Median (IQR) | 0 (0–2) | 0 (0–1) | 0 (0–4) |
| Camp location | | | |
| Teknaf | 455 (13.2) | 245 (12.9) | 210 (13.6) |
| Ukhiya | 2991 (86.8) | 1661 (87.1) | 1330 (86.4) |
| Smoking (n=2392)† | 513 (21.4) | 103 (7.2) | 410 (42.8) |
| BCG vaccination | 2974 (86.3) | 1665 (87.4) | 1309 (85.0) |
| Any relevant symptoms‡ | 1274 (37.0) | 723 (37.9) | 551 (35.8) |
| Any comorbidity§ | 430 (12.5) | 268 (14.1) | 162 (10.5) |

*Among those aged ≥6 years.
†Among those age ≥18 years.
‡Symptoms include fever (20.3%), cough (20.2%), dyspnoea (3.1%), anosmia (1.9%), chills (1.6%), fatigue (10.3%), sore throat (3.9%), headache (10.9%), ageusia (4.5%), nausea (3.9%), vomiting (3.9%), diarrhoea (5.8%), wheezing (1.4%), chest pain (2.9%), rhinorrhoea (11.8%), conjunctivitis (1.1%), other respiratory (0.6%), muscle aches (5.4%), myalgia (3.9%), rash (1.8%), loss appetite (3.7%), nosebleed (0.1%), seizures (0.2%).
§Comorbidities include cancer (0.1%), diabetes (2.5%), hypertension (5.8%), heart diseases (1.3%), asthma (1.9%), chronic lung disease (0.5%), chronic liver diseases (0.9%), thalassaemia or other chronic haematological disorder (0.1%), chronic kidney diseases (0.3%), chronic neurological impairment (0.3%), bone marrow diseases (0.5%).
FDMN, Forcibly Displaced Myanmar Nationals.

## Seroprevalence of SARS-CoV-2

Overall, 2090 (928 male and 1162 female) of 3446 study participants tested positive for SARS-CoV-2 antibody with the Wantai kit (table 2). The overall crude, weighted and weighted and test adjusted seroprevalence of SARS-CoV-2 antibodies was 60.7% (95% CI 59.0% to 62.3%), 57.4% (95% CI 55.2% to 59.7%) and 48.3% (95% CI 45.3% to 51.4%), respectively. The distribution of infection (weighted and Kantaro test adjusted) was 57.4 (95% CI 54.2 to 60.6) and did not differ between males and females. Children and adolescents (age 1–17 years) had lower prevalence (38.6%; 95% CI 33.8% to 43.4%) compared with the adults (58.1%; 95% CI 55.2% to 61.1%). There was no significant difference in infection between Teknaf and Ukhiya upazila. Non-smoker had significantly higher prevalence (60.5%; 95% CI 57.3% to 63.7%) compared with smoker (50.4%; 95% CI 43.9% to 56.6%). Almost half (45.7%, 95% CI 41.9% to 49.5%) of the asymptomatic participants had positive antibody tests. Participants with at least one comorbidity had higher seroprevalence than those with no comorbidity (57.8%; 95% CI 50.4% to 64.5% vs 47.2%; 95% CI 43.9% to 50.4%) (table 2). This distribution of seroprevalence across covariates were not different between males and females, except camp location where it was significantly higher among males in Ukhiya, lower among smoker males, higher among symptomatic females and higher among females who reported at least one comorbidity (online supplemental table 2).

The prevalence of SARS-CoV-2 antibody was higher in adults than in children, irrespective of their self-report of symptoms (figure 2). However, the prevalence was almost the same in two groups of adults (18–49 years and 50–94 years). Among those having comorbid conditions, the highest seroprevalence was found among participants with self-reported chronic liver disease (67.8%), followed by those with bone marrow disease (66.7%), hypertension (60.6%), diabetes (58.9%), heart disease patients (57.1%) and chronic lung disease/ asthma (52.5%) (figure 3). An increasing seroprevalence was observed with the number of self-reported comorbidities, 55.1% with one comorbidity and 71.8% with three or more comorbidities.

## Factors associated with anti-SARS-CoV-2 antibody seropositivity

Univariate logistic regression analysis identified increasing age (OR 1.02; 95% CI 1.02 to 1.03), increasing education level (OR 1.07; 95% CI 1.03 to 1.11), history of relevant symptoms (OR 1.28; 95% CI 1.06 to 1.54) and presence

**Table 2** Crude, weighted and test adjusted over weighted prevalence (95% CI) of SARS-CoV-2 antibody by sex across covariates, FDMN population, Bangladesh December 2020 (N=3446)

| Variable | Count Positive/total | Unweighted % (95% CI) | Weighted* % (95% CI) | Weight and test adjusted† % (95% CI) |
|---|---|---|---|---|
| **Age, years** | | | | |
| All | 2090/3446 | 60.7 (59.0 to 62.3) | 57.4 (55.2 to 59.7) | 48.3 (45.3 to 51.4) |
| 1–17 | 547/1054 | 51.9 (48.8 to 55.0) | 49.4 (45.6 to 53.2) | 38.6 (33.8 to 43.4) |
| 18–94 | 1543/2392 | 64.5 (62.6 to 66.4)‡ | 65.5 (63.2 to 67.8)‡ | 58.1 (55.2 to 61.1)‡ |
| **Sex** | | | | |
| Female | 1162/1906 | 61.0 (58.7 to 63.2) | 57.4 (54.2 to 60.6) | 48.3 (44.2 to 52.3) |
| Male | 928/1540 | 60.3 (57.8 to 62.7) | 57.4 (54.2 to 60.6) | 48.3 (44.2 to 52.3) |
| **Education§ (n=3160)** | | | | |
| No formal education | 1147/1847 | 62.1 (59.8 to 64.3) | 61.7 (58.8 to 64.5) | 53.5 (49.8 to 57.1) |
| Any education | 842/1313 | 64.1 (61.5 to 66.7) | 61.8 (58.3 to 65.2) | 53.6 (49.2 to 57.9) |
| **Camp location** | | | | |
| Teknaf | 261/455 | 57.4 (52.7 to 62.0) | 54.8 (48.9 to 60.6) | 45.1 (37.8 to 52.2) |
| Ukhiya | 1829/2991 | 61.2 (59.4 to 62.9) | 58.0 (55.5 to 60.4) | 49.0 (45.8 to 52.2) |
| **Smoking¶ (n=2392)** | | | | |
| No | 1240/1879 | 66.0 (63.8 to 68.1) | 67.5 (64.9 to 70.0) | 60.5 (57.3 to 63.7) |
| Yes | 303/513 | 59.1 (54.7 to 63.4)‡ | 59.1 (53.9 to 64.2)‡ | 50.4 (43.9 to 56.6)‡ |
| **BCG vaccination** | | | | |
| No | 306/472 | 64.8 (60.3 to 69.1) | 58.8 (52.7 to 64.7) | 50.0 (42.5 to 57.3) |
| Yes | 1784/2974 | 60.0 (58.2 to 61.8)‡ | 57.2 (54.7 to 59.6) | 48.0 (44.8 to 51.2) |
| **Any relevant symptoms*** | | | | |
| No symptom | 1282/2172 | 59.0 (56.9 to 61.1) | 55.3 (52.3 to 58.2) | 45.7 (41.9 to 49.5) |
| Any symptom | 808/1274 | 63.4 (60.7 to 66.1)‡ | 61.2 (57.7 to 64.6)‡ | 52.9 (48.5 to 57.2)‡ |
| **Any comorbidity††** | | | | |
| No comorbidity | 1800/3016 | 59.7 (57.9 to 61.4) | 56.5 (54.0 to 58.9) | 47.2 (43.9 to 50.4) |
| Any comorbidity | 290/430 | 67.4 (62.8 to 71.9)‡ | 65.2 (59.2 to 70.7)‡ | 57.8 (50.4 to 64.5)‡ |

*Weighted to FDMN population size (n=8 60 697) and distribution published by Refugee Relief and Repatriation Commissioner and United Nations High Commissioner for Refuge, 2020.
†Rogan-Gladen formula was used to adjust seroprevalence estimates to account for test sensitivity and specificity of Wantai kits validated by Kantaro.[27]
‡P<0.05 for comparisons between seropositives and categorical factors.
§Among those age ≥6 years.
¶Among those age ≥18 years.
**Symptoms include fever, cough, dyspnoea, anosmia, chills, fatigue, sore throat, headache, ageusia, nausea, vomiting, diarrhoea, wheezing, chest pain, rhinorrhoea, conjunctivitis, other respiratory, muscle aches, myalgia, rash, loss of appetite, nosebleed, seizures.
††Comorbidities include cancer, diabetes, hypertension, heart disease, asthma, chronic lung disease, chronic liver disease, thalassaemia or other chronic haematological disorder, chronic kidney disease, chronic neurological impairment, bone marrow disease.
FDMN, Forcibly Displaced Myanmar National.

of co-morbid conditions (OR 1.44; 95% CI 1.10 to 1.90) as the factors associated with higher risk of seropositivity, whereas smoking (OR 0.70; 95% CI 0.55 to 0.89) was associated with lower risk of seropositivity. In multiple logistic regression for all subjects adjusted for age, sex, camp locations, BCG vaccination, history of symptoms and comorbid conditions, the association for higher risk of seropositivity persisted for increasing age (aOR 1.01; 95% CI 1.01 to 1.02) and higher education (aOR 1.09; 95% CI 1.04 to 1.13) only. However, after adding smoking to the model for

adults (aged 18–94 years because smoking data were not collected for children), respondents who smoked had a lower risk of infection with SARS-CoV-2 (aOR 0.63, 95% CI 0.48 to 0.83). In addition, respondents who had a history of one or more symptoms had higher odds (aOR 1.34; 95% CI 1.08 to −1.67) of having the infection (table 3).

## DISCUSSIONS

We report a high seroprevalence of antibodies (total immunoglobulin including both IgG and IgM) to SARS-CoV-2

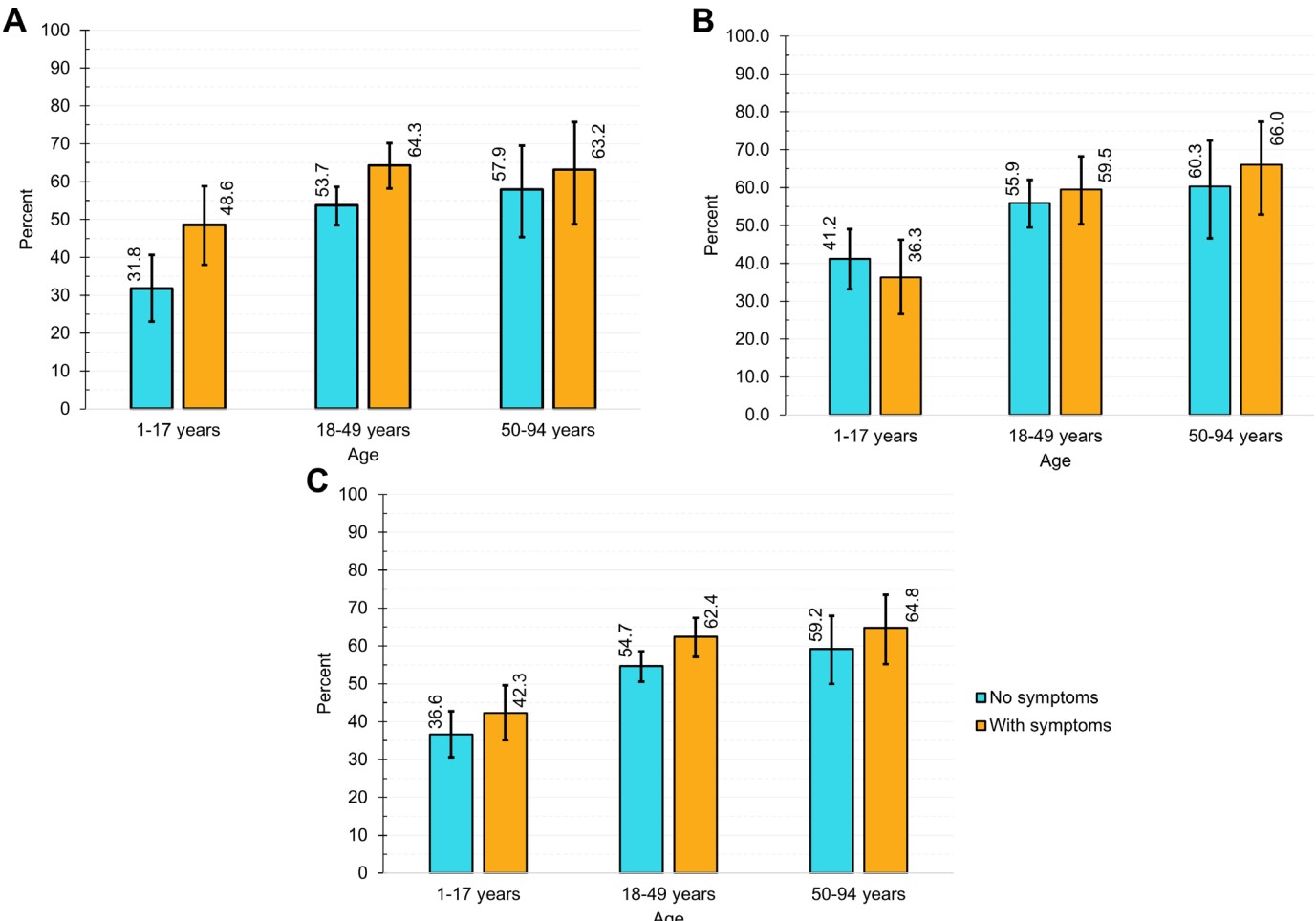

**Figure 2** Age-specific weighted test adjusted seroprevalence in male and female participants with or without history of symptoms, FDMN population, Bangladesh December 2020 (N=3446): (A) female, (B) male and (C) all subjects. Error bars indicate 95% CI. Weighted to FDMN population size (N=860 697) and distribution published by Refugee Relief and Repatriation Commissioner and United Nations High Commissioner for Refuge, 2020. Rogan-Gladen formula was used to adjust seroprevalence estimates to account for test sensitivity and specificity of Wantai kits validated by Kantaro.[25] FDMN, Forcibly Displaced Myanmar Nationals.

virus (48.3%) using a cross-sectional survey conducted in December 2020 among FDMNs living in Cox's Bazar district of Bangladesh. This high seroprevalence contrasts with a very low number of infections detected by RT-PCR before the study period. This probably is not surprising as other studies had similar observations even with low seroprevalence, like nationally representative study in Sierra Leone in March 2021 reported weighted seroprevalence of only 2.6%, but it was 43 times higher than the reported number of cases.[29] Seroprevalence study in Pakistan during October to November 2020 although reported a lower seroprevalence of 7.1% based on a low sensitive rapid antibody testing ('Bioperfectus' kits for IgG/IgM), the estimated seroprevalence was 62 times higher than the reported number of cases before the survey period in the sampled districts.[30] High seroprevalence of SARS-CoV-2 IgG (39.2%) was reported from metropolis of the Brazilian Amazon from October 2020 to February 2021, which was also 10 times higher than the case reported by the health authority as of December 2020.[31]

Although we found a higher seroprevalence (58.1%) in adults than children aged 1–17 years (38.6%), there was no difference in seroprevalence for older adults (aged 60 and above years) compared with middle-aged adults (aged 40–59 years), which supports the findings of the systematic review by Levin et al.[32] No difference of seroprevalence between sexes was found through our study. Earlier study in Bangladesh reported that females had a higher rate of both IgG and IgM seropositivity compared with males, but the findings were not significantly different.[13] Similar higher seropositivity among females was reported in Sierra Leone, without statistically significant difference.[29] Seroprevalence study in Pakistan also did not report any significant different results among males and females.[30] Seroprevalence study in South Korea during September 2020 to December 2020 also did not report any differences for sex or age group.[33]

Nearly half 45.7% of the people who did not report any symptom since March 2020 also had evidence of SARS-CoV-2 virus infection. They might have had very mild

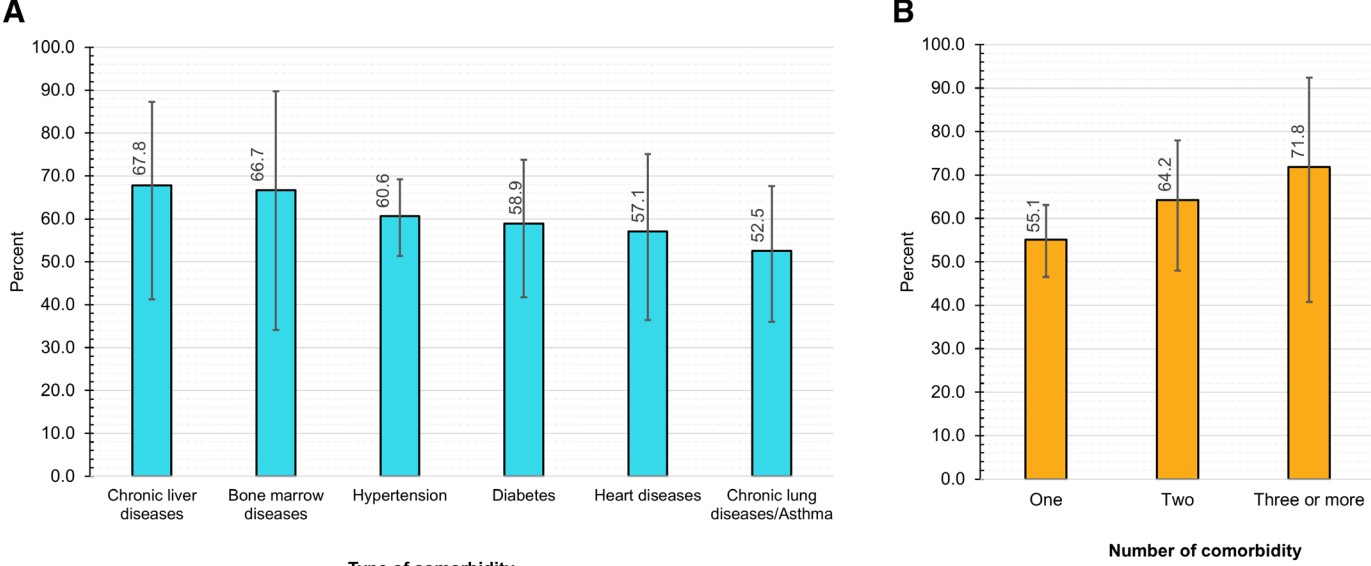

**Figure 3** Weighted test adjusted seroprevalence of SARS-CoV-2 among self-reported comorbid conditions, FDMN population, Bangladesh December 2020 (N=3446): (A) type of comorbidity, (B) number of comorbidities. Error bars indicate 95% CI. Weighted to FDMN population size (N=860 697) and distribution published by Refugee Relief and Repatriation Commissioner and United Nations High Commissioner for Refuge, 2020. Rogan-Gladen formula was used to adjust seroprevalence estimates to account for test sensitivity and specificity of Wantai kits validated by Kantaro.[25] FDMN, Forcibly Displaced Myanmar Nationals.

symptoms to recall correctly or were in fact asymptomatic. The younger age profile of the FDMN population also might explain the higher proportion of mild or asymptomatic infection.[34] Because this study was conducted during a pandemic, the time of the survey, geographical locations, age and sex composition of the sample, and type of antibody assessment should be considered before judging the findings or comparing it with other studies.

This study examines SARS-CoV-2 infections among those temporarily or chronically experiencing homelessness (or otherwise living in precarious housing) during the COVID-19 pandemic. The FDMN community has intrinsic differences from the host community, especially regarding awareness of COVID-19 issues and access to related information. Although their access to healthcare was beyond the scope of this survey, understanding the risk and needs of this group is crucial to containing the spread of the pandemic.

Our study findings indicated that by December 2020, almost half of the FDMN people were exposed to the SARS-CoV-2 infections, when population vaccination for COVID-19 yet to be started in the country. Another study in Sitakunda subdistrict in the Chattogram division (Cox's Bazar is also located in this division) of Bangladesh conducted in April 2021 has found 63.1% seroprevalence, which was conducted later than our study so expected to have more cumulative seropositive population and sample size, geographical context and type of antibody assessment were different than the current study.[35] The capital city of Dhaka also has a very high population density. Another study[13] reported a seroprevalence of IgG of 35.4% in the non-slum population of Dhaka city between April and October 2020. This study, on the other hand, reported 63.5% seroprevalence in slum areas of Dhaka city. Therefore, the high prevalence we found in FDMN is not unusual in the Bangladesh context. A seroprevalence study conducted in eight large slums of Mumbai, India in June–July 2020 reported a seroprevalence of 54.1%. However, the same study reported 16.1% seroprevalence in the non-slum areas of Mumbai.[36] A high seroprevalence of IgG antibodies was also reported in three high density communities in Harare, Zimbabwe during February to April 2021 (53.0%)[37] and in urban slums of India during December 2020 to January 2021 (34.7%),[38] both of which might have similar crowded living settings as of FDMN camps.

Like others,[6 39] we have observed a lower seroprevalence in children than adults. During the pandemic, children did not go to schools or any other places for an occupation to acquire the infection. Although most adult women stayed at home (because homemaking was their predominant occupation), they had similar infection rates as their male counterparts. Women might have acquired infections at household levels from their husbands or other male members, who might have exposure due to contact with people outside their homes. Imam/priests and incentive volunteers had the highest seroprevalence rates in our series (online supplemental figure 2), similar to other studies.[40] Our results also revealed that non-smokers are likely to have higher seroprevalence than smokers. Contrary to the putative effects of smoking on human health, smoking might have suppressed a due immune response to SARS-CoV-2 infections as reported by few other studies,[41–44] although examining this association was beyond the scope of our study. In addition, conflicting associations of BCG vaccination

**Table 3** Risk factors of weighted SARS-CoV-2 infection identified in univariate and multiple weighted logistic regression analysis, FDMN population, Bangladesh December 2020 (N=3446)

| Factors | Unadjusted | Adjusted | | |
| --- | --- | --- | --- | --- |
| | | Overall | (Age 1–17 years) | (Age 18–94 years) |
| | OR (95% CI) | OR (95% CI) | OR (95% CI) | OR (95% CI) |
| Age (continuous) | 1.02 (1.02 to 1.03)* | 1.01 (1.01 to 1.02)* | 1.09 (1.03 to 1.15)* | 1.01 (0.99 to 1.01) |
| Sex | | | | |
| Female | 1 | 1 | 1 | 1 |
| Male | 1.00 (0.83 to 1.20) | 1.00 (0.81 to 1.23) | 1.30 (0.89 to 1.88) | 1.05 (0.82 to 1.34) |
| Education (continuous) | 1.07 (1.03 to 1.11)* | 1.09 (1.04 to 1.13)* | 1.08 (0.97 to 1.21) | 1.06 (1.02 to 1.11)* |
| Camp location | | | | |
| Teknaf | 1 | 1 | 1 | 1 |
| Ukhiya | 1.14 (0.88 to 1.47) | 1.10 (0.83 to 1.46) | 1.12 (0.68 to 1.84) | 1.07 (0.78 to 1.47) |
| Smoking | | | | |
| No | 1 | | | 1 |
| Yes | 0.70 (0.55 to 0.89)* | | | 0.63 (0.48 to 0.83)* |
| BCG vaccination | | | | |
| No | 1 | 1 | 1 | 1 |
| Yes | 0.94 (0.72 to 1.22) | 0.96 (0.72 to 1.28) | 1.18 (0.55 to 2.53) | 0.85 (0.66 to 1.12) |
| Symptoms† | | | | |
| No | 1 | 1 | 1 | 1 |
| Yes | 1.28 (1.06 to 1.54)* | 1.20 (0.99 to 1.47) | 1.12 (0.77 to 1.63) | 1.34 (1.08 to 1.67)* |
| Comorbidity‡ | | | | |
| No comorbidity | 1 | 1 | 1 | 1 |
| One or more | 1.44 (1.10 to 1.90)* | 1.10 (0.82 to 1.48) | 1.06 (0.54 to 2.11) | 1.16 (0.85 to 1.60) |

*P<0.05.
†Symptoms include fever, cough, dyspnoea, anosmia, chills, fatigue, sore throat, headache, ageusia, nausea, vomiting, diarrhoea, wheezing, chest pain, rhinorrhoea, conjunctivitis, other respiratory, muscle aches, myalgia, rash, loss appetite, nosebleed, seizures.
‡Comorbidities include cancer, diabetes, hypertension, heart disease, asthma, chronic lung disease, chronic liver disease, thalassaemia or other chronic haematological disorder, chronic kidney disease, chronic neurological impairment, bone marrow disease.
FDMNs, Forcibly Displaced Myanmar Nationals.

with SARS-CoV-2 infections have been reported. We did not observe any significant association with BCG vaccination. The proposition of a better cellular response in those who have BCG vaccination[45–47] is beyond the scope of our study.

Our study among FDMN population had several limitations. The studies relying on antibody response have their inherent limitations. The estimates reported here might be affected by the waning antibody levels over time. However, it should be noted that vaccination for SARS-CoV-2 had not been initiated in Bangladesh (and hence among the study population) at the time of the study, and therefore was not likely to influence the seroprevalence estimates. The household roster could be completed for about 71% of targeted subjects (online supplemental table 1) because the listing of households was done several months before the survey. Non-response at the household level was higher than expected (13.6%). This might have caused overestimation of the seroprevalence, as reflected by the unweighted seroprevalence result in table 2. The

population and household level weighting should address the issue of non-response. Again, the Wantai ELISA might have cross-reactivity with antibodies to other common non-SARS-CoV-2 human coronaviruses, which are known to circulate worldwide continuously,[48 49] leading to a false-positive result in some instances. Although we were not able to check for cross-reactivity against other coronaviruses, we adjusted seroprevalence estimates to account for test sensitivity and specificity of Wantai validated by Kantaro to address issue of cross-reactivity from past infection specially during prepandemic period from any non-SARS-CoV-2 human corona viruses. Earlier study in Kenya showed slightly lower specificity of Wantai compared with in-house (KWTRP) IgG antibody ELISA kit for prepandemic samples, indicating likely overestimation of population seroprevalence.[50] Furthermore, self-reports on comorbid conditions and probable COVID-19 symptoms due to a long recall period might impact our study results. In addition, as the study protocol has been adopted from WHO unity study, we did not include questions on

non-pharmaceutical public health measures, for example, use of masks which can influence the population seroprevalence as reported by other studies in Bangladesh.[51]

## CONCLUSIONS

The findings of this study imply that the prevalence of seropositivity is likely to be much higher than the presumed RT-PCR confirmed rates in FDMN population. A high seroprevalence across age and sex groups, irrespective of a history of symptoms, may indicate an inadequacy of infection control measures. Because of a very high population density in the camps, social distancing may not work. Other well-known public health measures, such as face masks and mass vaccination, are warranted.

**Author affiliations**
[1]Epidemiology, Institute of Epidemiology Disease Control and Research, Dhaka, Bangladesh
[2]Virology, Institute of Epidemiology Disease Control and Research, Dhaka, Bangladesh
[3]WHO Emergency Sub-Office, World Health Organization, Cox's Bazar, Bangladesh
[4]Research and Publication, World Health Organization Bangladesh, Dhaka, Bangladesh
[5]Co-ordination Center, Ministry of Health and Family Welfare, Cox's Bazar, Bangladesh
[6]IEDCR Field Laboratory, World Health Organization, Cox's Bazar, Bangladesh
[7]Zoonosis, Institute of Epidemiology Disease Control and Research, Dhaka, Bangladesh
[8]Director, Institute of Epidemiology Disease Control and Research, Dhaka, Bangladesh
[9]Additional Director General, Directorate General of Health Services, Dhaka, Bangladesh

**Acknowledgements** We gratefully acknowledge the field team, MOHFW, DGHS, divisional coordinators, civil surgeons, Upazila health and family planning officers and health assistants for their support. We are also thankful to the following local and international NGO and development partners: WHO, RRRC, UNHCR; Relief International; FH/MTI; ICRC/BDRCS; BRAC; icddr,b; World Bank; Bureau of Population, Refugees and Migration—US Department of State through Driving Public Health Action in the World's Largest Refugee Camps; Government of Norway. We are indebted to the camp community leaders (Majhis) to facilitate the data collection and FDMN camp residents for their participation with the study.

**Contributors** MR, ESE, MMZ, TS and MSF conceptualised this study. MR, ASMA, DSK, FH, ESE, ANA, MMZ, TS and MSF developed the methodology and study design. MR, SRK, DSK, FH, ESE, NA and MMZ wrote the protocol. MR, SRK, DSK, FH, ESE, NA, MMB and MKA developed the data collection tool. DSK and RB developed the software used. ANA, MMZ, TS and MSF validated analyses, data and methodology used. MR, DSK, MSI and RB conducted the formal analysis. MR, SRK, ASMA, DSK, FH, ANA, NA, SS, DP, RB, RI, AM, MS, MMB, ARS, MHK, MMZ and MSF undertook the investigation. ASMA, ANA, KvH, MMZ and TS provided resources for the study. MR, DSK and MSI curated the data. MR, DSK, FH, MSI and MMZ wrote the original draft of the manuscript. MSI designed the visualisations used. MR, SRK, ASMA, FH, ANA, NA, DP, RI, AM, MMB, KvH, MMZ, TS and MSF provided supervision. MR, SRK, ASMA, DSK, FH, NA, DP, AM, MS, ARS, MKA, KvH, MMZ, TS and MSF administered this project. MMZ, TS and MSF acquired the funding for the study. All authors participated in reviewing and editing the manuscript. TS acting as the guarantor.

**Funding** This work was supported by the WHO (Ref. No. 2020/1060317-2).

**Disclaimer** However, no fund has been used to prepare this manuscript.

**Map disclaimer** The inclusion of any map (including the depiction of any boundaries therein), or of any geographic or locational reference, does not imply the expression of any opinion whatsoever on the part of BMJ concerning the legal status of any country, territory, jurisdiction or area or of its authorities. Any such expression remains solely that of the relevant source and is not endorsed by BMJ. Maps are provided without any warranty of any kind, either express or implied.

**Competing interests** None declared.

**Patient and public involvement** Patients and/or the public were involved in the design, or conduct, or reporting, or dissemination plans of this research. Refer to the Methods section for further details.

**Patient consent for publication** Not applicable.

**Ethics approval** This study involves human participants and was approved by Institutional Review Board of the Institute of Epidemiology, Disease Control and Research (Approval Memo No. IEDCR/IRB/2020/24). Participants gave informed consent to participate in the study before taking part.

**Provenance and peer review** Not commissioned; externally peer reviewed.

**Data availability statement** Data are available on reasonable request. To protect patients' privacy, the ethical approval obtained for this study prevents human data from being disclosed publicly. Requests to access the datasets should be directed to Professor Tahmina Shirin, Director, IEDCR, email: director@iedcr.gov.bd; directoriedcr@gmail.com.

**ORCID iDs**
Mahbubur Rahman http://orcid.org/0000-0001-8577-8281
Ferdous Hakim http://orcid.org/0000-0003-2376-3978
Nawroz Afreen http://orcid.org/0000-0003-4633-4754
Md Sahidul Islam http://orcid.org/0000-0001-6361-2870
Mohammad Mostafa Zaman http://orcid.org/0000-0002-1736-1342

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
