## [Reviewer comments · BMJ Open]

ARTICLE DETAILS

TITLE (PROVISIONAL)	Seroprevalence of SARS-CoV-2 antibodies among Forcibly Displaced Myanmar Nationals in Cox's Bazar, Bangladesh 2020: a population-based cross-sectional study
AUTHORS	Rahman , Mahbubur; Khan, Samsad; Alamgir, A. S. M; Kennedy, David; Hakim, Ferdous; Evers, Egmond; Afreen, Nawroz; Alam, Ahmed; Islam, Md. Sahidul; Paul, Debashish; Bhuiyan, Rijwan; Islam, Raisul; Moureen, Adneen; Salimuzzaman, M; Billah, Mallick; Sharif, Ahmed; Akter, Mst. Khaleda; Sultana, Sharmin; Khan, Manjur; von Harbou, Kai; Zaman, MM; Shirin, Tahmina; Flora, Meerjady Sabrina

VERSION 1 – REVIEW

REVIEWER	Vallinoto, Antonio Universidade Federal do Pará, Instituto de Ciências Biológicas, Laboratório de Virologia
REVIEW RETURNED	29-Jul-2022

GENERAL COMMENTS	The study of Rahman et al. brings a very interesting results about the seroprevalence of anti-SARS-CoV-2 IgG antibodies amon among Forcibly Displaced Myanmar Nationals in Cox's Bazar, Bangladesh. It is a very large epidemiological study and very well done! However, it is no clear enough why the authors used first an ELISA kit to detect total antibodies and then a kit to detect IgG, aiming to assess the validation (sensibility and specificity). I suggest the authors to compare their results with those reported in other communities around the world, for example, those reported by da Silva Torres MK, et al. Seroprevalence and risk factors for COVID-19 in the metropolis of the Brazilian Amazon. Sci Rep. 2022 May 20;12(1):8571. doi: 10.1038/s41598-022-12629-z. Were there vaccinated individuals among the cohort investigated? It was not clear. The ethnic differences were not reported by the authors. Was it considered in the data analysis? How could it impact the results? Would It be a limitation of the study? How was the mask wearing among the investigated subjects? It can influence the seroprevalence.
--

REVIEWER	Lorenz, Eva
-----------------	-------------

	Bernhard Nocht Institute of Tropical Medicine, Infectious Disease Epidemiology
REVIEW RETURNED	19-Aug-2022

GENERAL COMMENTS	Thank you for the opportunity to review this interesting article on SARS-CoV-2 seroprevalence and associated factors among the Forcibly Displaced Myanmar Nationals in Bangladesh based on data from a two-stage cross-sectional study in December 2020. This work will be of interest in describing an important vulnerable population of will help to understand both the progression of the epidemic and in particular associated risk factors. The article presents results from a very well designed and impressive study. I have few comments regarding the manuscript and recommend acceptance after minor revisions on this manuscript. Strength and limitations of the study In the fourth bullet point, a higher-than-expected non-response rate at the household level is stated. Please extend this point with regard to how (direction and extent) this might have impacted results. This also applies to the Discussion section where this aspect is mentioned but again not discussed. Introduction 2nd paragraph: l.40ff: A word seems to be missing in 'Bangladesh currently hosts around than One million...'. 'than' should be 'than'. Results Factors associated with anti-SARS-CoV-2 antibody seropositivity: I recommend removing the wording 'statistically significant' and just state that these factors are associated. It has the same strength of evidence in the context of such exploratory analyses and avoids confusing it with a confirmatory study set up and powered to specifically assess risk factor associations. Conclusions: The conclusions have a strong focus on control measures such as social distancing and other public health measures. To support these, the reader would need to understand what has concretely been the situation in the study population. It would be good to add respective information in a dedicated paragraph in the Methods/Setting section.
--

VERSION 1 – AUTHOR RESPONSE

Reviewer #1		
1	The study of Rahman et al. brings a very interesting results about the seroprevalence of anti-SARS-CoV-2 IgG antibodies among Forcibly Displaced Myanmar Nationals in Cox's	No response needed, thank you for your positive comment and critical review.

	Bazar, Bangladesh. It is a very large epidemiological study and very well done!	
2	It is not clear enough why the authors used first an ELISA kit to detect total antibodies and then a kit to detect IgG, aiming to assess the validation (sensitivity and specificity).	The Wantai ELISA might have cross-reactivity with antibodies to other common human coronaviruses leading to a false-positive result in some instances. Although we were not able to check for cross-reactivity against other coronaviruses, we adjusted seroprevalence estimates to account for test sensitivity and specificity of Wantai validated by Kantaro to address issue of cross-reactivity from past infection from non-SARS-CoV-2 human coronavirus, which we have reported in discussion limitation section. We specifically used a SARS-CoV-2 IgG antibody test kit (Kantaro), to exclude past infection specially during pre-pandemic period from any non-SARS-CoV-2 human coronavirus. Earlier study showed slightly lower specificity of Wantai compared with in-house (KWTRP) IgG antibody ELISA for pre-pandemic samples. We have updated the earlier text along with relevant references for further clarification. This now reads as (Line 417-426): '...Wantai ELISA might have cross-reactivity with antibodies to other common non-SARS-CoV-2 human coronaviruses, which are known to circulate worldwide continuously,^{48,49} leading to a false-positive result in some instances. Although we were not able to check for cross-reactivity against other coronaviruses, we adjusted seroprevalence estimates to account for test sensitivity and specificity of Wantai validated by Kantaro to address issue of cross-reactivity from past infection specially during pre-pandemic period from any non-SARS-CoV-2 human coronavirus. Earlier study in Kenya showed slightly lower specificity of Wantai compared with in-house (KWTRP) IgG antibody ELISA kit for pre-pandemic samples, indicating likely overestimation of population seroprevalence.⁵⁰ References added: 48. Ogimi C, Kim YJ, Martin ET, et al. What's New With the Old Coronaviruses? J Pediatric Infect Dis Soc 2020;9(2):210-17. doi: 10.1093/jpids/piaa037 [published Online First: 2020/04/22] 49. Grimwood K, Lambert SB, Ware RS. Endemic Non-SARS CoV-2 Human Coronaviruses in a Community-Based Australian Birth Cohort. Pediatrics 2020;146(5) doi: 10.1542/peds.2020-009316 50. Nyagwange J, Kutima B, Mwai K, et al. Comparative performance of WANTAI ELISA for total immunoglobulin to receptor binding protein and an ELISA for IgG to spike protein in detecting SARS-CoV-2 antibodies in Kenyan populations. Journal of Clinical Virology 2022;146:105061. doi: https://doi.org/10.1016/j.jcv.2021.105061 1

3	I suggest the authors to compare their results with those reported in other communities around the world, for example, those reported by da Silva Torres MK, et al. Seroprevalence and risk factors for COVID-19 in the metropolis of the Brazilian Amazon. Sci Rep. 2022 May 20;12(1):8571. doi: 10.1038/s41598-022-12629-z.	Thank you for the suggestion. To compare the results with those reported in other communities around the world, we have added few studies in discussion section including the suggested article. This now reads as (Line 340-343): ‘High seroprevalence of SARS-CoV-2 IgG (39.2%) was reported from metropolis of the Brazilian Amazon from October 2020 to February 2021, which was also 10 times higher than the case reported by the health authority as of December 2020.³¹’ Again, a sentence added in Line 353-355: ‘Seroprevalence study in South Korea during September 2020 to December 2020 also did not report any differences for sex or age group.³³’ And, in Line 385-388: ‘A high seroprevalence of IgG antibodies was also reported in three high density communities in Harare, Zimbabwe during February to April 2021(53.0%)³⁷ and in urban slums of India during December 2020 to January 2021 (34.7%),³⁸ both of which might have similar crowded living settings as of FDMN camps.’ References added: 31. da Silva Torres MK, Lopes FT, de Lima ACR, et al. Seroprevalence and risk factors for COVID-19 in the metropolis of the Brazilian Amazon. Scientific reports 2022;12(1):8571. doi: 10.1038/s41598-022-12629-z [published Online First: 2022/05/21] 33. Nah E-H, Cho S, Park H, et al. Nationwide seroprevalence of antibodies to SARS-CoV-2 in asymptomatic population in South Korea: a cross-sectional study. BMJ open 2021;11(4):e049837. doi: 10.1136/bmjopen-2021-049837 37. Fryatt A, Simms V, Bandason T, et al. Community SARS-CoV-2 seroprevalence before and after the second wave of SARS-CoV-2 infection in Harare, Zimbabwe. EClinicalMedicine 2021;41 doi: 10.1016/j.eclinm.2021.101172 38. Murhekar MV, Bhatnagar T, Thangaraj JWV, et al. SARS-CoV-2 seroprevalence among the general population and healthcare workers in India, December 2020–January 2021. International Journal of Infectious Diseases 2021;108:145-55. doi: https://doi.org/10.1016/j.ijid.2021.05.040
4	Were there vaccinated individuals among the cohort investigated?	Our study was conducted in December 2020. While Bangladesh began the administration of COVID-19 vaccines on 27 January 2021 and mass vaccination started on 7 February 2021. The vaccination programme for FDMN was started on 10 August 2021. Therefore the study was completed before the vaccination programme was launched in the FDMN population and Bangladesh, which was reported in discussion section as “it should be noted that vaccination for SARS-CoV-2 had not been initiated in Bangladesh at the time of the study, and therefore was not likely to influence the seroprevalence estimates”. We have updated the text, which now reads as (Line 409-410):

		'....vaccination for SARS-CoV-2 had not been initiated in Bangladesh (and hence among the study population) at the time of the study, and therefore was not likely to influence the seroprevalence estimates.'
5	The ethnic differences were not reported by the authors. Was it considered in the data analysis? How could it impact the results? Would It be a limitation of the study?	FDMN itself a whole ethnic group commonly known as 'Rohingyas' in the Rakhine State of Myanmar (REF). There are no other ethnicities are present among FDMN population. We have added this information in the text with reference which reads as (Line 120-122): 'Bangladesh currently hosts around one million FDMNs (commonly known as 'Rohingyas', an ethnic group in the Rakhine State of Myanmar who are forcibly displaced in Bangladesh)¹⁵.....' Reference added: 15. Uddin Md Zahed I, Jenkins B. The Politics of Rohingya Ethnicity: Understanding the Debates on Rohingya in Myanmar. Journal of Muslim Minority Affairs 2022;42(1):117-35. doi: 10.1080/13602004.2022.2064054
6	How was the mask wearing among the investigated subjects? It can influence the seroprevalence	Thank you for raising this important issue. Indeed, practice of mask use can influence the seroprevalence result. But, as the study protocol has been adopted from WHO unity study, we did not include questions on mask use as this was not considered as our main objective and there were several other studies available addressing this. But we have included this issue in discussion section as a limitation. This now reads as (Line 427-430) 'Additionally, as the study protocol has been adopted from WHO unity study, we did not include questions on non-pharmaceutical public health measures, e.g. use of mask which can influence the population seroprevalence as reported by other study in Bangladesh.⁵¹ Reference added: 51. Raqib R, Sarker P, Akhtar E, et al. Seroprevalence of SARS-CoV-2 infection and associated factors among Bangladeshi slum and non-slum dwellers in pre-COVID-19 vaccination era: October 2020 to February 2021. PloS one 2022;17(5):e0268093. doi: 10.1371/journal.pone.0268093.
Reviewer #2		

1	Thank you for the opportunity to review this interesting article on SARS-CoV-2 seroprevalence and associated factors among the Forcibly Displaced Myanmar Nationals in Bangladesh based on data from a two-stage cross-sectional study in December 2020. This work will be of interest in describing an important vulnerable population of will help to understand both the progression of the epidemic and in particular associated risk factors. The article presents results from a very well designed and impressive study. I have few comments regarding the manuscript and recommend acceptance after minor revisions on this manuscript.	No response needed, thank you for your positive comment and critical review.
---	--	---

2	Strength and limitations of the study In the fourth bullet point, a higher-than-expected non-response rate at the household level is stated. Please extend this point with regard to how (direction and extent) this might have impacted results. This also applies to the Discussion section where this aspect is mentioned but again not discussed.	Thank you for your suggestion. We have updated fourth bullet point of the strength and limitations of the study section. This now reads as (Line 91-92): 'This might have caused overestimation of the seroprevalence, which was addressed by population and household level weighting.' As suggested, discussion section also have been updated to elaborate possible impact of non-response rate. This now reads as (Line 414-418): 'This might have caused overestimation of the seroprevalence, as reflected by the unweighted seroprevalence result in table 2.'
3	Introduction 2nd paragraph: l.40ff: A word seems to be missing in 'Bangladesh currently hosts around than One million...'. 	Thank you very much for this careful observation. The text has been updated and this now reads as (Line 120-121): 'Bangladesh currently hosts around one million FDMNs....'
4	Results Factors associated with anti-SARS-CoV-2 antibody seropositivity: I recommend removing the wording 'statistically significant' and just state that these factors are associated. It has the same strength of evidence in the context of such exploratory analyses and avoids confusing it with a confirmatory study set up and powered to specifically assess risk factor associations.	Thank you very much for the justified suggestion. We have revised the text by removing the word 'statistically significant'. This now reads as (Line 313-317): 'Univariate logistic regression analysis identified increasing age (OR: 1.02; 95% CI: 1.02–1.03), increasing education level (OR: 1.07; 95% CI: 1.03–1.11), history of relevant symptoms (OR: 1.28; 95% CI: 1.06–1.54) and presence of co-morbid conditions (OR: 1.44; 95% CI: 1.10–1.90) as the factors associated with higher risk of seropositivity....'

5	Conclusions The conclusions have a strong focus on control measures such as social distancing and other public health measures. To support these, the reader would need to understand what has concretely been the situation in the study population. It would be good to add respective information in a dedicated paragraph in the Methods/Setting section.	The objective of the study was to assess the status of seroprevalence of SARS-CoV-2 in FDMN population at the time of the study. The behavioral aspects of the Population in line with control of infection with SARS-CoV-2 are beyond the scope of this study, as per design of the WHO unity study. However, we have updated Methods/Setting section to reflect high dense living situation in FDMN camps. This reads now as (Line 140-142): ‘All 34 FDMN camps of Cox's Bazar district were included (Supplemental Figure 1), which present with a population density of 40,000 people per square kilometre on average with some places reaching 70,000.²⁰’ Reference added: Inter Sector Coordination Group. Joint Multi-Sector Needs Assessment (J-MSNA), Rohingya Refugees, Cox's Bazar, Bangladesh 2020.
---	--	---